# Inflammatory Brain Lesions as Omen of Primary Central Nervous System Lymphoma: A Case Report and Literature Review

**DOI:** 10.3390/brainsci11020191

**Published:** 2021-02-04

**Authors:** Yeong Jin Kim, Seul Kee Kim, Tae-Young Jung, In-Young Kim, Kyung-Hwa Lee, Kyung-Sub Moon

**Affiliations:** 1Department of Neurosurgery, Chonnam National University Research Institute of Medical Science, Chonnam National University Hwasun Hospital and Medical School, Hwasun 58128, Korea; mbdosa88@naver.com (Y.J.K.); jung-ty@jnu.ac.kr (T.-Y.J.); kiy87@hanmail.net (I.-Y.K.); 2Department of Radiology, Chonnam National University Research Institute of Medical Science, Chonnam National University Hwasun Hospital and Medical School, Hwasun 58128, Korea; kimsk.rad@gmail.com; 3Department of Pathology, Chonnam National University Research Institute of Medical Science, Chonnam National University Hwasun Hospital and Medical School, Hwasun 58128, Korea

**Keywords:** primary central nervous system lymphoma, tumor regression, diagnosis, inflammation, tumefactive lesion

## Abstract

We report a rare case that was initially diagnosed as an inflammatory lesion and ultimately confirmed as primary central nervous system lymphoma (PCNSL) in an immunocompetent patient who was not treated with corticosteroid prior to the initial biopsy. A 70-year-old female patient presented with numbness in the left side of face, arm, and leg. Brain magnetic resonance imaging (MRI) revealed a lesion with intense gadolinium (Gd)-enhancement in the ventral portion of the midbrain. A stereotactic biopsy demonstrated mixed T-cell and B-cell infiltrating inflammatory lesions without demyelination. Three months after postoperative treatment with steroid, the lesion markedly decreased on follow-up MRI. Twenty-six months after the initial attack, she complained of dysarthria and urinary incontinence. Repetitive MRI showed a lesion with homogeneous enhancement, extensively involving the bilateral cerebral hemisphere, corpus callosum, and the right middle cerebellar peduncle. The confirmed diagnosis was diffuse large B-cell lymphoma on the second biopsy. Despite our best efforts, she died 38 months after disease onset. Based on review of the literature and our case, preceding inflammatory lesions are not always demyelinating and T-cell dominant inflammatory lesions. When the initial biopsy reveals an inflammatory lesion in an old-aged patient, the clinician should keep in mind the development of PCNSL and perform close clinical and radiological observations for a timely diagnosis.

## 1. Introduction

Primary central nervous system lymphoma (PCNSL) is a rare brain tumor, accounting for 3% of all brain tumors and 6% of all extra-nodal lymphomas [1]. In recent decades, the incidence of PCNSL has increased in patients aged > 65 years. PCNSL can develop in patients of any age, with the highest incidence rate in the fifth to seventh decade. Although PCNSL is more common in immunosuppressed patients, its incidence has also increased in immunocompetent individuals [2]. The pathogenesis of PCNSL in immunocompetent individual is still poorly understood. There are a few case studies demonstrating inflammatory brain lesions preceding the development of PCNSL. In previous reports, preceding inflammatory lesions, also called “sentinel lesions”, are described as demyelinating, T-cell dominant inflammatory lesions, and PCNSL can develop in chronological order in different locations [3,4,5]. The relationship between preceding inflammatory lesions and pathogenesis of PCNSL has not yet been revealed. Elucidating this relationship may be helpful to understand the development of PCNSL in immunocompetent patients. 

Median survival of patients has improved since the introduction of methotrexate-based chemotherapy for PCNSL treatment [6]. Therefore, timely diagnosis and treatment are important to improve the prognosis of patients with PCNSL [1]. Herein, we report a rare case of PCNSL with a preceding inflammatory lesion in an immunocompetent patient who was not treated with a corticosteroid prior to initial biopsy.

## 2. Case Presentation

A 70-year-old woman without previous medical history presented with a one-week history of numbness in the left side of face, arm, and leg. A neurologic examination revealed no additional neurologic sign except above-mentioned symptoms. There were no abnormal laboratory findings from blood samples. Brain MRI showed a contrast-enhancing lesion in the ventral portion of the right midbrain with perilesional edema (Figure 1A–C). 

With suspect of glioma or lymphoma, Leksell frame-based stereotactic biopsy was performed without cerebrospinal fluid (CSF) examination. Biopsy of the lesion showed lymphocytic infiltration composed of mixed CD3+ T-cells and CD20+ B-cells with low Ki-67 index in the perivascular space and glial stroma. Demyelinating features such as foamy macrophages engulfing myelin debris were not detected (Figure 2). Descriptive diagnosis was made as an inflammatory lesion with dense lymphocytic infiltration. 

Brain computed tomography (CT) and MRI were performed after the biopsy procedure. Proper specimens were sampled with minimal hemorrhage (Figure 1D). After careful consideration, the patient was treated with intravenous methylprednisolone 160 mg daily for 5 days, leading to subsidence of symptoms gradually. Subsequently, daily doses of 10 mg oral prednisolone were introduced for remaining mild symptoms. Three months after the initial brain MRI, there was nearly complete resolution of the lesion. Brain MRI was performed at 6-month interval and did not reveal specific pathologic findings. Twenty-six months after the initial attack, she suddenly developed urinary incontinence, quadriparesis, and mental deepening. Brain MRI showed homogenously enhancing lesions in both cerebral hemispheres and a curvilinear enhancing lesion involving the middle cerebellar peduncle, distant from the initial lesion (Figure 3A–C).

Neuro-navigation guided burr hole biopsy targeting the right frontal lobe revealed diffuse large B-cell lymphoma (Figure 4). A subsequent systemic evaluation provided no evidence of a systemic lymphoma. She was treated with methotrexate chemotherapy and cytarabine chemotherapy due to methotrexate-induced diabetic insipidus. This treatment provided clinical improvement and nearly complete remission of the lesions on subsequent brain MRI (Figure 3D). Thirty-five months after the initial attack, she suddenly developed right hemiparesis and cognitive impairment. Brain MRI showed homogenously enhancing lesions in the left basal ganglia and pons along the corticospinal tract. She refused further treatment and died at 38 months after the disease onset.

## 3. Discussion

We present a patient with a steroid-unaffected inflammatory brain lesion preceding PCNSL. This rare clinical presentation implies that tumefactive inflammatory brain lesions, not only demyelinating lesions, could be omen of PCNSL development and that close clinical observation is mandatory for such patients. PCNSL can be associated with preceding inflammatory lesions as summarized in a few case reports [3,7]. In rare cases, preceding inflammatory lesions resolved spontaneously or through steroid treatment, ultimately resulting in PCNSL development.

It is difficult to distinguish imaging features of space-occupying inflammatory lesions and brain tumors like PCNSL and glioma from the beginning. Although several imaging studies have been performed, they have not clearly contributed to differential diagnosis. MRI features such as T1 hypo-intensity, T2 hyper-intensity, and contrast enhancing are common to that disease. With the metabolic data from positron emission tomography (PET), advanced MRI techniques (diffusion, spectroscopy, or perfusion) may be useful for differential diagnosis and disease monitoring [8,9]. After all, histological confirmation is required for final diagnosis as done in our case. Unlike other brain tumors, resection of PCNSL is not recommended. Infiltrative spreading and the deep seated location impede tumor removal. Chemotherapy-sensitive, radiotherapy-sensitive, and corticosteroid-responsive features of PCNSL make aggressive surgical resection meaningless. Stereotactic biopsy or other less invasive technique is recommended when CSF examination and eye examination are unable to reach a diagnosis [6]. In the present case, MR images suggested brain tumors like glioma or PCNSL in the deep seated midbrain. We inevitably performed Leksell frame-based stereotactic biopsy. 

Most previous studies have reported that preceding inflammatory lesions were histopathologically characterized by a predominance of T cell infiltration with few B cell and demyelination [4,10]. Demyelination is inconclusive, but a characteristic feature to help narrow diagnosis candidates, including infectious, metabolic, toxic and autoimmune etiology [11]. Especially in patients treated with inappropriate steroid treatment before pathologic confirmation, incomplete demyelination with extensive inflammation at the initial biopsy can raise the possibility of PCNSL in recurred brain lesions [11]. Unlike previous studies, our case did not show demyelinating features like foamy macrophages engulfing myelin debris. In addition, mixed T- and B- lymphocytes infiltrated in the perivascular space and glial stroma. Not only our case, but also a few previous case studies have reported minimal inflammation and a mixture of T- and B-cells without demyelination [3,12]. Thus, T-cell dominant lymphocytic infiltration and demyelinating features are not mandatory characteristics of heralding inflammatory lesions. 

There may be questions about the accuracy of the biopsy result. Corticosteroid treatment before initial biopsy can mask clinical symptoms, imaging appearance, and even histological findings of PCNSL. Corticosteroids have a cytolytic effect on lymphoma cells, making them disappear and causing inaccurate diagnosis [13]. As we thought of PCNSL as a disease candidate, corticosteroid treatment was not applied before the initial biopsy in the present case. The stereotactic biopsy target site might obscure histopathological diagnosis. The specimen acquired by stereotactic biopsy could be an appropriate target tissue or an uncertain surrounding tissue. Even with appropriately targeted tissues, indefinite diagnosis can be made because of heterogenous pathology of PCNSL [13]. In our case, brain CT and MRI performed after stereotactic biopsy supported that the lesion was sampled properly with minimal hemorrhage.

The interval between preceding inflammatory lesions and following PCNSL ranged from 6 months to 4 years [7]. The preceding inflammatory lesions can disappear spontaneously or by corticosteroid treatment without showing clues of PCNSL. Some reports have proposed clinical criteria to raise suspicion for preceding inflammatory lesions and PCNSL. However, there are many exceptions [10]. The present case had descriptive diagnosis as an inflammatory lesion with dense lymphocytic infiltration based on predominant lymphocytic population composed of mixed T- and B- cells with low Ki-67 labeling index in the perivascular space and glial stroma without mitoses, necrosis, or microvascular proliferation. Her initial symptoms and imaging features disappeared after two months of low dose corticosteroid treatment. After that, the patient was free of symptoms and the inflammatory lesion resolved completely for 23 months. At that time, we could not think that PCNSL would awfully develop because her symptoms were mild and the lesion responded well to corticosteroid for a long time. Twenty-five months after the initial histopathological diagnosis, her newly emerging symptoms suddenly exacerbated within three days. Repetitive biopsy from another site confirmed diffuse large B cell lymphoma, different from previous findings. Furthermore, we ruled out the possibility of Epstein-Barr virus-positive PCNSL due to high dose steroid treatment by negative result of in situ hybridization for Epstein-Barr virus encoded RNA performed on the second biopsy. 

PCNSL might emerge unexpectedly due to abnormal lymphocyte proliferation after independent preceding inflammatory lesions. It can be triggered by preceding inflammatory lesions in pro-inflammatory patients [4]. PCNSL associated with chronic inflammation has also been reported in immunocompetent patients [14]. On the other hand, preceding inflammatory lesions may be the first immunological defense against emerging PCNSL, causing diagnostic difficulty [5]. Similar cases have been reported repeatedly, but the pathogenetic association between preceding inflammatory lesion and PCNSL is yet mere hypothetical, and further studies on the underlying mechanism are needed. Although the underlying mechanism is unknown, inflammatory response plays an important role in developing PCNSL. Histopathological features as in the present case may be helpful for understanding the pathogenesis of PCNSL in immunocompetent patients. To find common features to give clues, we searched previous studies in English literature that reported biopsy-proven preceding inflammatory lesions and following PCNSL in immunocompetent and corticosteroid-untreated patients before initial biopsy [3,4,5,15,16,17] (Table 1). Combining six cases from the literature and our case, we found that patient’s age was the most noticeable factor. PCNSL in immunocompetent patients usually developed between the age of 55–70 years, while inflammatory demyelinating disease without following PCNSL was present in young adulthood [18]. Similar to PCNSL, preceding inflammatory lesions in the reported cases developed between the age of 44–70 years. Other clinical characteristics were not unified. In brain imaging, either space-occupying lesions or not-occupying lesions such as chronic lymphocytic inflammation with pontine perivascular enhancement responsive to steroids (CLIPPERS) could be preceding inflammatory lesions [17]. In microscopic findings, inflammatory lesions could be preceding lesions regardless of demyelination or T-cell dominant infiltration. Even steroid responsiveness was variable among previous reports. Collectively, inflammatory brain lesions in aged patients should be suspected as preceding lesions of PCNSL.

Anderson et al. have named inflammatory demyelinating brain lesions preceding PCNSL as “sentinel lesions” [3]. After that, reported studies have used “sentinel lesions” to describe inflammatory brain lesions preceding PCNSL. However, the term of “sentinel lesions” is not suitable to describe chronologically related diseases. The relationship between “sentinel lesions” and PCNSL resembles transient ischemic attack and ischemic stroke rather than sentinel lymph node and breast cancer. Thus, we propose that inflammatory brain lesions preceding PCNSL should be termed as “omen” or “prodromal lesion” of PCNSL.

## 4. Conclusions

In conclusion, preceding inflammatory lesions are not always demyelinating lesions or T-cell dominant inflammatory lesions. Inflammatory brain lesions in old-aged patients have a possibility to be an “omen” or “prodromal lesion” of PCNSL. Thus, clinicians must carry out close clinical and radiological observations to reach a timely diagnosis. It is important not to hesitate repetitive biopsy for suspicious lesions.

## Figures and Tables

**Figure 1 brainsci-11-00191-f001:**
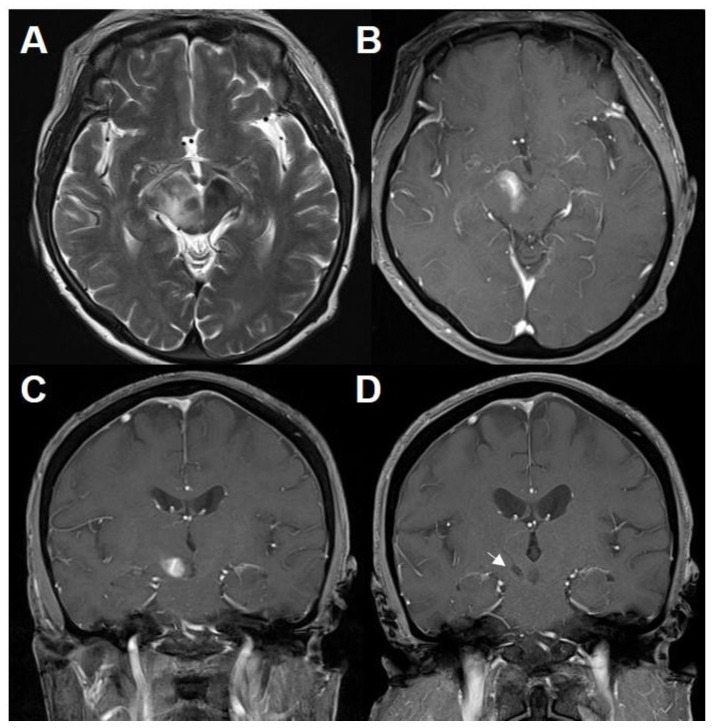
MRI of a preceding inflammatory lesion. T2-weighted (**A**) and T1-weighted MRI with gadolinium (**B**,**C**) showing an enhancing lesion in the ventral portion of the right midbrain with perilesional edema. Follow-up T1-weighted MRI with gadolinium (**D**) after steroid therapy demonstrating the proper location of biopsy (white arrow).

**Figure 2 brainsci-11-00191-f002:**
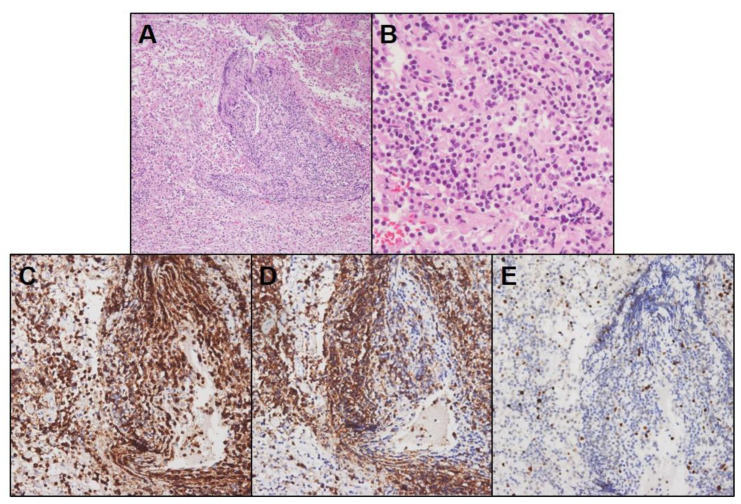
Pathological images of the preceding inflammatory lesion. Histopathological examination of the specimen (**A**,**B**) showing dense mononuclear cell infiltration in the perivascular space and glial stroma without necrosis, mitosis, or microvascular proliferation (hematoxylin and eosin staining, 100×, 400×). Immunohistochemical staining showing mixed CD3+ (**C**) and CD20+ (**D**) lymphocytic population with a low Ki-67 index (**E**) (200×).

**Figure 3 brainsci-11-00191-f003:**
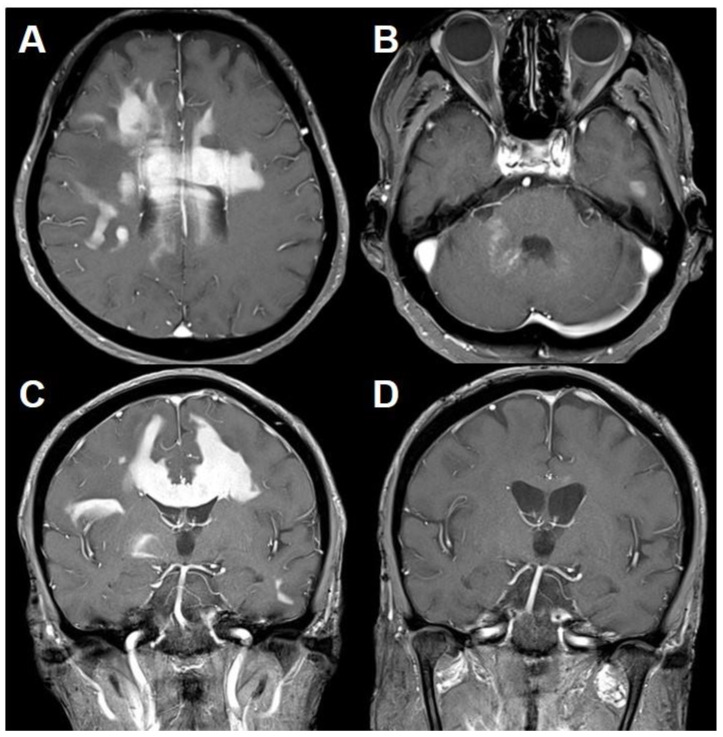
MRI of primary central nervous system lymphoma (PCNSL) following the preceding inflammatory lesion. T1-weighted MRI with gadolinium (**A**–**C**) showing homogeneously enhancing lesions in the bilateral cerebral hemisphere and corpus callosum with punctate enhancing lesions in the right middle cerebellar peduncle. Follow-up T1-weighted MRI with gadolinium (**D**) demonstrating complete resolution of the lesions after chemotherapy therapy.

**Figure 4 brainsci-11-00191-f004:**
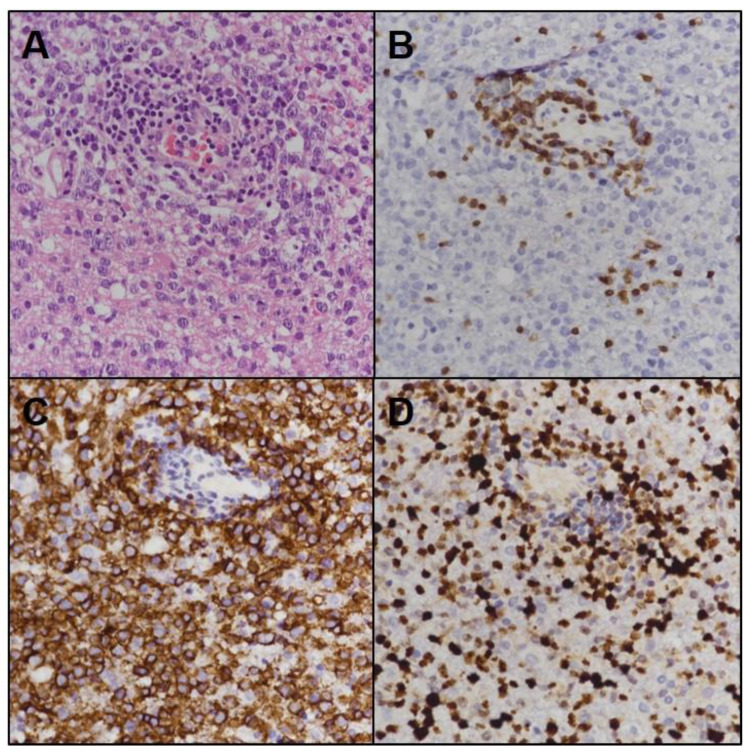
Pathological images of PCNSL. Histopathological examination of the specimen (**A**) showing atypical, large, and irregular cells with vesicular nuclei and prominent nucleoli (hematoxylin and eosin staining, 200×). The majority of tumor cells were negative for CD3 (**B**) but positive for CD20 (**C**) with a high Ki-67 index (**D**) (200×).

**Table 1 brainsci-11-00191-t001:** Summary of PCNSL with preceding inflammatory lesions in immunocompetent and corticosteroid-untreated patients before initial biopsy.

Case No */Ref.	Age/Sex	First Lesions				Interval ^#^ (mos)	Relapsed Lesions				
		Symptom	MRI	Pathology	Tx		Symptom	MRI	Pathology	Tx	Px ^!^
1/[15]	56/F	Fatigue	Small multifocal T2-hyperintense lesion	Demyelination with lymphocyte infiltration, reactive astrocytosis	Corticosteroid	5	Dementia Quadriplegia	Widespread T2-hyperintense white matter lesion	PCNSL (B-cell) with severe T-cell in filtration	None (autopsy)	13 mos survival
2/[3]	49/F	Ataxia Deafness	Enhancing lesion	Normal tissue with minimal inflammation	Corticosteroid	11	Confusion Dysarthria	Enhancing lesion	PCNSL (B-cell)	Unknown	Unknown
3/[4]	59/F	Facial paralysis Hemihypesthesia	T2-hyperintense, mild enhancing lesion	T-cell dominant inflammatory lesion with partial demyelination	Corticosteroid	28	Dysmetria Ataxia	Enhancing lesions	PCNSL (B-cell)	MTX-chemotherapy; Stem cell apheresis; WBRT	34 mos survival
4/[17]	58/M	Ataxia Diplopia	Punctate, curvilinear enhancing lesions	T-cell dominant inflammatory lesion without demyelination	Corticosteroid & Azathioprine	14	Ataxia Diplopia	Punctate, curvilinear enhancing lesions	PCNSL (B-cell)	None	37 mos survival
5/[16]	70/M	Dementia Hemiparesis	Homogenous and ring enhancing lesions	Demyelinating changes with reactive astrocytes	Corticosteroid	3	Dementia Hemiparesis	Homogenous enhancing lesion	PCNSL (B-cell)	WBRT	8 mos survival
6/[5]	44/F	Hand numbness Ataxia Vertigo	Irregular enhancing lesion	Both myelin and axonal loss with focal demyelination	Corticosteroid & MTX	10	Dementia Seizure Mental change	Enhancing lesions	PCNSL (B-cell)	None (autopsy)	10 mos survival
7 (current case)	70/F	Facial numbness Hemiparesthesia	Homogenous enhancing lesion	Inflammatory lesion with mixed T/B-cell infiltration without demyelination	Corticosteroid	26	Quadriparesis Mental change	Homogenous enhancing lesions	PCNSL (B-cell)	MTX-chemotherapy	38 mos survival

*: in the order of publication date, ^#:^ Interval: from 1st biopsy to 2nd biopsy, ^!^: survival duration from initial diagnosis; MRI; magnetic resonance imaging, Tx; treatment, Px; prognosis, PCNSL; primary central nervous system lymphoma, MTX; methotrexate, WBRT; whole brain radiation treatment, mos; months, F; female, M; male.

## Data Availability

Not applicable.

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
