# Peer review of "Inflammatory Brain Lesions as Omen of Primary Central Nervous System Lymphoma: A Case Report and Literature Review"

_brainsci, 2021, doi:10.3390/brainsci11020191_

Round 1
Reviewer 1 Report
Interesting case. Please expand on the discussion and add at least 3-4 more references. There are minor syntax errors and typos that need correction.
Good images and case.
This paper can be improved by the following:
Additional references that can be helpful to include:
1) Primary CNS Lymphoma. Christian Grommes et al
2) Diagnostic red flags: steroid‐treated malignant CNS lymphoma mimicking autoimmune inflammatory demyelination by Alonso Barrantes‐Freer et al
3) Primary CNS Lymphomas: Challenges in Diagnosis and Monitoring by Chiavazza et al
The following can be included in discussion/limitations:
1) The authors state “We report a rare case that was initially diagnosed as inflammatory lesions and ultimately 18 confirmed as primary central nervous system lymphoma (PCNSL) in an immunocompetent patient who was not treated with corticosteroid.”
However, the patient developed primary CNS lymphoma after the initial treatment with steroids to the initial lesion/pathology. Therefore, it is not appropriate to rule out steroids as a possible contributing factor.
2) Molecular analysis or clonality studies were not conducted initially on the inflammatory lesion to rule out CNS Lymphoma.
3) Possibility of sampling bias with the biopsy which the authors have confirmed as “accurate”.
Author Response
I’d like to thank the editor and reviewers of ‘Brain Sciences’ for taking time to review our article (brainsci-1079164). I have made some corrections and clarifications in the manuscript after going over your comments. The changes are summarized below. (In the revised manuscript, edited sentences are highlighted by the 'track changes' function of MS word):
[ Reviewer 1]
Interesting case. Please expand on the discussion and add at least 3-4 more references. There are minor syntax errors and typos that need correction.
Good images and case.
This paper can be improved by the following:
- Additional references that can be helpful to include:
1) Primary CNS Lymphoma. Christian Grommes et al
Response) We have added the citation, as the reviewer recommended.
2) Diagnostic red flags: steroid‐treated malignant CNS lymphoma mimicking autoimmune inflammatory demyelination by Alonso Barrantes‐Freer et al
Response) The reference was already cited in the 1st draft. We have inserted an additional sentence to strengthen the point of the citation in the discussion, as follows: “Especially in patients treated with inappropriate steroid treatment before pathologic confirmation, incomplete demyelination with extensive inflammation at the initial biopsy can raise the possibility of PCNSL in recurred brain lesions.” (line 146-148 in revised paper)
3) Primary CNS Lymphomas: Challenges in Diagnosis and Monitoring by Chiavazza et al
Response) We have changed the sentence and included this reference in the discussion, as the reviewer recommended: “With the metabolic data from positron emission tomography (PET), advanced MRI techniques (diffusion, spectroscopy, or perfusion) may be useful for differential diagnosis and disease monitoring.” (line 131-133 in revised paper)
The following can be included in discussion/limitations:
1) The authors state “We report a rare case that was initially diagnosed as inflammatory lesions and ultimately confirmed as primary central nervous system lymphoma (PCNSL) in an immunocompetent patient who was not treated with corticosteroid.”
However, the patient developed primary CNS lymphoma after the initial treatment with steroids to the initial lesion/pathology. Therefore, it is not appropriate to rule out steroids as a possible contributing factor.
Response) With the original sentence, we wanted to emphasize that the patient was not treated with steroid prior to the initial biopsy, which might cause extensive degeneration and could result in inconclusive pathology. To clarify the point, we have changed the sentences in the abstract as follows: “We report a rare case that was initially diagnosed as inflammatory lesions and ultimately confirmed as primary central nervous system lymphoma (PCNSL) in an immunocompetent patient who was not treated with corticosteroid prior to the initial biopsy.” and “Three months after postoperative treatment with steroid, the lesion markedly decreased on follow-up MRI.” (abstract in revised paper))
2) Molecular analysis or clonality studies were not conducted initially on the inflammatory lesion to rule out CNS Lymphoma.
Response) Although our hospital performs molecular examinations such as immunoglobulin heavy chain (IgH) gene rearrangement or immunoglobulin κ light chain (IgK) gene rearrangement analysis to check the clonality of malignant lymphoma, we could not get the gene arrangement analysis done since the initial biopsy was too small to extract enough DNA considering the amount of stereotactic specimens. The hematopathologist of our hospital also agreed that the size of initial biopsy was inadequately small for clonality study.
3) Possibility of sampling bias with the biopsy which the authors have confirmed as “accurate”.
Response) We performed the initial biopsy at the right-side of the midbrain. To exclude the possibility of sampling errors, we took brain imaging of the patient after the stereotactic biopsy. Repeating biopsy at the eloquent area of the midbrain was burdensome, and the patient’s symptoms were gradually improved with steroid treatment after the initial biopsy. As shown in the follow-up coronal MR image (Fig. 1D), the biopsy was considered to be performed properly at the epicenter of the lesion.
To clarify these points, we have deleted the sentence “Thus, there was no doubt that the biopsy was accurate.” We have also fixed the sentence “In our case, brain CT & MRI performed after stereotactic biopsy confirmed that proper lesions were biopsied with minimal hemorrhage.” into “In our case, brain CT & MRI performed after the stereotactic biopsy supported that the lesion was sampled properly with minimal hemorrhage.” (line 164-165 in revised paper). And we have added sentence “Even with appropriately targeted tissues, indefinite diagnosis can be made because of heterogenous pathology of PCNSL.” with a new citation (line 162-164 in revised paper).
Reviewer 2 Report
The authors present a case of an elderly patient with a focal inflammatory lesion of the midbrain who developed multifocal diffuse large B-cell lymphoma of the CNS over 2 years later, despite absence of overt immunodeficiency or prolonged therapy. The report is well written, and the images are of good quality. Nonetheless, the presented conclusion that the patient's terminal DLBCL arose from an indolent preceding inflammatory lesion suffers from several serious flaws.
1) The authors claim that the initial brainstem biopsy was "accurate" on the basis of post-operative imaging, but this claim ignores the possibility of sampling bias, which is often the case with stereotactic biopsy specimens. Primary CNS lymphomas can exhibit striking regional heterogeneity, with non-neoplastic inflammatory infiltrates present immediately adjacent to tumor foci. Our institution has unfortunately cared for a number of such patients, who required multiple biopsy specimens to achieve accurate diagnosis.
2) The possibility that the initial "inflammatory" lesion was a low-grade B-cell lymphoma was not explored via IgH clonality studies or other molecular analysis, especially given the unusually high proportion of CD20+ B-cells in the photomicrograph (Fig. 2D). While primary low-grade B-cell lymphoproliferative disorders of the CNS are rare, evolution of an existing neoplasm is more likely than malignant transformation of an inflammatory disorder of unclear etiology.
3) Similarly, the patient's presentation and lack of demyelination on initial biopsy does not suggest a longstanding, chronic inflammatory process that is often required to produce lymphoma, such as is the case for other inflammation associated lymphomas with prostheses, pyothorax, etc.
4) Lastly, in situ hybridization for EBER should be performed or shown for the DLBCL specimen, as it's also possible that the patient developed EBV+ lymphoma following administration of high dose steroids to treat the initial brainstem pathology.
Author Response
I’d like to thank the editor and reviewers of ‘Brain Sciences’ for taking time to review our article (brainsci-1079164). I have made some corrections and clarifications in the manuscript after going over your comments. The changes are summarized below. (In the revised manuscript, edited sentences are highlighted by the 'track changes' function of MS word):
[ Reviewer 2]
The authors present a case of an elderly patient with a focal inflammatory lesion of the midbrain who developed multifocal diffuse large B-cell lymphoma of the CNS over 2 years later, despite absence of overt immunodeficiency or prolonged therapy. The report is well written, and the images are of good quality. Nonetheless, the presented conclusion that the patient's terminal DLBCL arose from an indolent preceding inflammatory lesion suffers from several serious flaws.
1) The authors claim that the initial brainstem biopsy was "accurate" on the basis of post-operative imaging, but this claim ignores the possibility of sampling bias, which is often the case with stereotactic biopsy specimens. Primary CNS lymphomas can exhibit striking regional heterogeneity, with non-neoplastic inflammatory infiltrates present immediately adjacent to tumor foci. Our institution has unfortunately cared for a number of such patients, who required multiple biopsy specimens to achieve accurate diagnosis.
Response) As you mentioned, multiple biopsies from multiple targets are commonly performed in our institution. However, the brainstem location of the initial lesion prevented us from performing aggressive biopsies. Repeating biopsy at the eloquent area of the midbrain was burdensome, and the patient’s symptoms were gradually improved with steroid treatment after the initial biopsy. As shown in the follow-up coronal MR image (Fig. 1D), the biopsy was considered to be performed properly at the epicenter of the lesion.
To clarify these points, we have deleted the sentence “Thus, there was no doubt that the biopsy was accurate.” We have also fixed the sentence “In our case, brain CT & MRI performed after stereotactic biopsy confirmed that proper lesions were biopsied with minimal hemorrhage.” into “In our case, brain CT & MRI performed after the stereotactic biopsy supported that the lesion was sampled properly with minimal hemorrhage.” (line 164-165 in revised paper). And we have added sentence “Even with appropriately targeted tissues, indefinite diagnosis can be made because of heterogenous pathology of PCNSL.” with a new citation (line 162-164 in revised paper).
2) The possibility that the initial "inflammatory" lesion was a low-grade B-cell lymphoma was not explored via IgH clonality studies or other molecular analysis, especially given the unusually high proportion of CD20+ B-cells in the photomicrograph (Fig. 2D). While primary low-grade B-cell lymphoproliferative disorders of the CNS are rare, evolution of an existing neoplasm is more likely than malignant transformation of an inflammatory disorder of unclear etiology.
Response) Although our hospital performs molecular examinations such as immunoglobulin heavy chain (IgH) gene rearrangement or immunoglobulin κ light chain (IgK) gene rearrangement analysis to check the clonality of malignant lymphoma, we could not get the gene arrangement analysis done since the initial biopsy was too small to extract enough DNA considering the amount of stereotactic specimens. The hematopathologist of our hospital also agreed that the size of initial biopsy was inadequately small for clonality study.
3) Similarly, the patient's presentation and lack of demyelination on initial biopsy does not suggest a longstanding, chronic inflammatory process that is often required to produce lymphoma, such as is the case for other inflammation associated lymphomas with prostheses, pyothorax, etc.
Response) We implied the hypothesis about the relationship between the initial inflammatory lesion and the subsequent lymphoma. The major hypothesis is ‘immunological defense against PCNSL’. Although the preceding lesion is not a longstanding lesion and is different from pyothorax-induced lymphoma, we focused that inflammatory lesion itself could be initiating factor to subsequent unknown tumorigenic mechanism.
And to emphasize that it is mere speculation, we have changed the sentence “PCNSL may emerge suddenly due to explosive abnormal proliferation of B lymphocytes.” into “PCNSL might emerge unexpectedly due to abnormal lymphocyte proliferation after independent preceding inflammatory lesion.” (line 185-186 in revised paper). And we have added a sentence “Similar cases have been reported repeatedly, but the pathogenetic association between preceding inflammatory lesion and PCNSL is yet mere hypothetical, and further studies on the underlying mechanism are needed.” (line 190-193 in revised paper)
4) Lastly, in situ hybridization for EBER should be performed or shown for the DLBCL specimen, as it's also possible that the patient developed EBV+ lymphoma following administration of high dose steroids to treat the initial brainstem pathology.
Response) We performed in situ hybridization exam for Epstein-Barr virus encoded RNA not on the initial biopsy sample, but on the second sample that was confirmed by DLBCL. To get rid of confusion, we have added the sentence “Furthermore, we ruled out the possibility of Epstein-Barr virus-positive PCNSL due to high dose steroid treatment by negative result of in situ hybridization for Epstein-Barr virus encoded RNA performed on the second biopsy.” in the discussion (line 181-184 in revised paper).
Reviewer 3 Report
It is an interesting and well-documented case report.
Major comment: The histopathological analysis of the initial lesion classified as inflammatory, and of the right frontal lobe appearing years later which was identified as PCNSL indicate that they truly represent two different pathological entities. In the discussion it should be clear that the possible pathogenetic association between the initial inflammatory lesion and the subsequent lymphoma is a mere speculation or hypothesis and that both processes could be independent entities not pathogenetically related.
Author Response
I’d like to thank the editor and reviewers of ‘Brain Sciences’ for taking time to review our article (brainsci-1079164). I have made some corrections and clarifications in the manuscript after going over your comments. The changes are summarized below. (In the revised manuscript, edited sentences are highlighted by the 'track changes' function of MS word):
[ Reviewer 3]
It is an interesting and well-documented case report.
Major comment: The histopathological analysis of the initial lesion classified as inflammatory, and of the right frontal lobe appearing years later which was identified as PCNSL indicate that they truly represent two different pathological entities. In the discussion it should be clear that the possible pathogenetic association between the initial inflammatory lesion and the subsequent lymphoma is a mere speculation or hypothesis and that both processes could be independent entities not pathogenetically related.
Response) To emphasize that it is mere speculation, we have changed the sentence “PCNSL may emerge suddenly due to explosive abnormal proliferation of B lymphocytes.” into “PCNSL might emerge unexpectedly due to abnormal lymphocyte proliferation after independent preceding inflammatory lesions.” (line 185-186 in revised paper). And we have added a sentence “Similar cases have been reported repeatedly, but the pathogenetic association between preceding inflammatory lesion and PCNSL is yet mere hypothetical, and further studies on the underlying mechanism are needed.” (line 190-193 in revised paper)
Round 2
Reviewer 2 Report
The authors have adequately addressed the raised concerns and modified the language of their conclusions appropriately.
Reviewer 3 Report
The manuscript has been significantly improved.